# Primary Metabolic Variations in Maize Plants Affected by Different Levels of Nitrogen Supply

**DOI:** 10.3390/metabo15080519

**Published:** 2025-08-01

**Authors:** The Ngoc Phuong Nguyen, Rose Nimoh Serwaa, Jwakyung Sung

**Affiliations:** Department of Crop Science, College of Agriculture, Life and Environment Sciences, Chungbuk National University, Cheongju 28644, Republic of Korea; phuongnguyen@chungbuk.ac.kr (T.N.P.N.); roseserwaanimoh47@chungbuk.ac.kr (R.N.S.)

**Keywords:** tissue-specific, nitrogen assimilation, carbon metabolism, maize cultivation

## Abstract

**Background/Objectives**: Nitrogen (N) is an essential macronutrient that strongly influences maize growth and metabolism. While many studies have focused on nitrogen responses during later developmental stages, early-stage physiological and metabolic responses remain less explored. This study investigated the effect of different nitrogen-deficient levels on maize seedling growth and primary metabolite profiles. **Methods**: Seedlings were treated with N-modified nutrient solution, which contained 0% to 120% of the standard nitrogen level (8.5 mM). **Results**: Nitrogen starvation (N0) significantly reduced plant height (by 11–14%), shoot fresh weight (over 30%) compared to the optimal N supply (N100). Total leaf nitrogen content under N0–N20 was less than half of that in N100, whereas moderate N deficiency resulted in moderate reductions in growth and nitrogen content. Metabolite analysis revealed that N deficiency induced the accumulation of soluble sugars and organic acids (up to threefold), while sufficient N promoted the synthesis of amino acids related to nitrogen assimilation and protein biosynthesis. Statistical analyses (PCA and ANOVA) showed that both genotypes (MB and TYC) and tissue type (upper vs. lower leaves) influenced the metabolic response to nitrogen, with MB displaying more consistent shifts and TYC exhibiting greater variability under moderate stress. **Conclusions**: These findings highlight the sensitivity of maize seedlings to early nitrogen deficiency, with severity influenced by nitrogen level, tissue-specific position, and genotype; thus underscore the close coordination between physiological growth and primary metabolic pathways in response to nitrogen availability. These findings expand current knowledge of nitrogen response mechanisms and offer practical insights for improving nitrogen use efficiency in maize cultivation.

## 1. Introduction

Nitrogen (N) is one of the most important mineral elements for plant growth and development [1,2]. As a primary macronutrient, nitrogen is essential for the synthesis of amino acids, the building block of proteins and nucleic acids such as DNA and RNA, which are important for regulating cell division and genetic expression [3]. Therefore, studying N in plants is important for understanding plant growth and development, improving plant yield and quality [4]. Low-N stress can seriously inhibit plant growth and reduce plant yield and quality. Nitrogen stress occurs when nitrogen availability is insufficient to meet the plant’s physiological demand, especially during rapid growth periods such as vegetative expansion and reproductive development. This stress affects both the morphological and biochemical processes of plants, leading to decreased yield and reduced biomass accumulation [2,5,6]. In maize seedlings, nitrogen limitation frequently contributes to expanded root architecture and reduced shoot biomass production as well as leaf development [7,8,9,10,11]. At the reproductive stage, inadequate nitrogen reduces kernel number per ear and kernel weight, limits assimilate supply for ear growth, and interferes with pollen development, therefore, decreasing overall reproduction efficiency [12].

Nitrogen stress alters primary metabolism in plants as nitrogen is a building block component of many primary metabolites, including amino acids, nucleotides, and chlorophyll. Under nitrogen-deficient conditions, plants reprogram their metabolic networks to adapt to reduced nitrogen availability, in which consistent metabolic profiles changes were recorded in many crops (maize, barley, rice, rapeseed, apple, and tomato) [13]. Amino acid plays a critical role in regulating plant growth, including involvement in protein synthesis, precursor for secondary metabolites, and compatible osmolytes. Under abiotic stress, amino acids can be utilized as an alternative respiratory substrate and metabolic signal to save energy [14]. Under sufficient nitrogen conditions, plants actively synthesize amino acids such as glutamine, glutamate, aspartate, and asparagine to support protein synthesis, while nitrogen assimilation-related amino acids—glutamine, alanine, glycine, and cysteine—also function as activators of the target of rapamycin (TOR) signaling pathway, which promotes growth by inducing amino acid synthesis and inhibiting autophagy and amino acid degradation [15]. In contrast, under N starvation conditions, most free amino acids were recorded to decrease in maize leaves and roots [9,16,17], as a result of limited supply of ammonium and carbon skeletons, subsequently resulting in retarded vegetation growth, reduced protein content, and decreased cell division [18].

On the other hand, organic acid contents were also reduced under N stress conditions, accompanied by amino acids, which then reflects in a reduction in the demand for carbon skeletons and limits the nitrogen assimilation activities [13,19]. The level of organic acid in the tricarboxylic acid (TCA) cycle, such as 2-oxoglutarat, malate, citrate, and fumarate reduced during nitrogen stress [5]. In contrast, nitrogen deficiency often results in the accumulation of soluble sugars such as glucose, fructose, and sucrose [9,13]. This occurs because photosynthesis continues even as nitrogen-limited growth slows down, leading to excess carbohydrate storage [20], and on the other hand, with limited nitrogen availability, carbon is translocated into energy storage compounds like sugar and starch [21,22]. In maize, limited nitrogen content reduced sink strength, like growing leaves, developing ears, and limiting sugar export from mature leaves to newly developing leaves [23,24].

The physical and mechanical responses of maize under nitrogen limitation vary depending on the timing of deficiency, developmental stage, and cultivation environment [25]. While many studies have focused on the vegetation and reproduction stages, few have investigated in early growth stage. Recent findings suggest that even short-term N deprivation at early stages can trigger significant shifts in metabolic activity, including alterations in nitrate uptake, amino acid metabolism, and carbon–nitrogen balance [15,26]. In particular, primary metabolites such as amino acids, organic acids, and carbohydrates serve not only as building blocks for growth but also as key indicators of the plant’s physiological condition under stress. Therefore, a better understanding of how maize seedlings respond to varying levels of nitrogen deficiency at the metabolic level is essential for developing strategies to enhance early-stage nitrogen use efficiency and stress resilience. The objective of this study is to examine the effect of different nitrogen-deficient levels on the growth and primary metabolites changes in maize seedlings, focusing on amino acid, organic acid, and carbohydrate contents. To minimize environmental variability, the experiment was conducted in greenhouse conditions with a hydroponic system.

## 2. Materials and Methods

### 2.1. Plant Growth and Total Nitrogen Content Analysis

Maize (*Zea mays* L.) cultivars used in this experiment were two Korean cultivars, Mibaek#2 (MB) and Taeyangchal (TYC). MB, developed in 2005 from inbred lines HW9 and HW3, is one of the most widely cultivated varieties in Korea due to its high yield and resistance to lodging, B.maydis, and E.turcicum. TYC, on the other hand, is a newly released variety developed by Chungbuk Province (South Korea). Maize seeds were germinated in plastic trays containing artificial soil (Baroker Sangto, Seoul Bio, Eumseong, Chungbuk, Republic of Korea), and placed under a natural photoperiod in the experimental greenhouse of Chungbuk National University, South Korea. After 10 days from sowing, the second leaf stage-seedlings were gently washed with water to remove the soil, then transferred to 5 L plastic containers (10 plants per container), each pot containing 0.5-strength nutrient solution. The nutrient solution was modified Hoagland solution containing Ca(NO_3_)_2_ 2.5 mM, KNO_3_ 2.5 mM, MgSO_4_ 1 mM, K_2_H_2_PO_4_ 0.25 mM, Fe-EDTA 0.03 mM, NH_4_NO_3_ 0.5 mM, H_3_BO_3_ 2 μM, MnCl_2_ 0.2 μM, ZnSO_4_ 0.19 μM, CuSO_4_ 0.01 μM, H_2_MoO_4_ 0.03 μM. At 15 days after transplanting, the nutrient solution was replaced with treatments differing in nitrogen concentration as 0 (N-deficiency), 20, 40, 60, 80, 100, and 120% of modified Hoagland solution (Table 1). The shortage in potassium and calcium supply was compensated with an equivalent of KCl and CaCl_2_. The solution pH was maintained between 5.8–6.0, and the nutrient solution was replaced every 7 days.

Maize plants (5 plants per treatment) at 15 days after N treatments were randomly taken to measure growth parameters. The shoot (leaves and stem) and root parts were separated, and then their length was measured using a ruler. The shoot length is counted to the top of the main plant stem. The root length is measured according to the longest root. The shoot and root fresh weight are measured by an electronic scale and then dried at 80 °C for 72 h until no changes in mass to measure the shoot and root dry weight, as described in [27].

To measure total nitrogen content, the first two leaves showing senescence were thus removed, and the remaining leaves were separated into 2 groups. Lower leaves (LL) contained the 3rd and 4th leaves, which are fully developed leaves, and upper leaves (UL) contained the 5th to 7th leaves as young and emerging leaves. Dry samples (0.5 g) from two leaf groups were taken and then blended into powder. Total nitrogen contents were measured with a C/N analyzer (VarioMax CN Analyzer, Elementar GmbH, Langenselbold, Germany).

### 2.2. Amino Acid, Organic Acid, and Carbohydrate Content Measurement

Depending on growth and total nitrogen content, three biological replicates at each selected treatment, N0, N60, and N100, were taken for metabolite measurement. Metabolite profiling was performed following a modified methanol extraction and derivatization protocol [28]. Approximately 30 mg of dried maize leaf tissue was extracted with 1.0 mL of 70% methanol, then filtered through a 0.2 µm syringe filter. The filtrate (200 µL) was transferred to a fresh 1.5 mL microtube and dried using a speed vacuum concentrator. The dried sample was dissolved in 50 µL of 20,000 ppm methyl hydroxyl chloride amine (MHCA) in pyridine and mixed with 50 µL of internal standard solution (1000 ppm fluoranthene). The mixture was incubated in a thermo mixer at 30 °C for 90 min, then 100 µL of BSTFA (N,O-Bis(trimethylsilyl) trifluoroacetamide) was added, vortexed again, and incubated at 60 °C for 30 min. The derivatized sample was transferred to a GC glass vial with an insert and subjected to gas chromatography–mass spectrometry (GC-MS) analysis. The GC-MS system consisted of a Thermo Scientific TSQ 8000 triple quadrupole mass spectrometer coupled to a Trace 1310 gas chromatograph (Thermo Fisher Scientific, Waltham, MA, USA). Separation was achieved using a 60 m 0.25 mm id 0.25 µm DB-5 column (Agilent Technologies, Santa Clara, CA, USA) with helium as the carrier gas. Spectral searching was carried out by consulting the NIST Mass Spectral Library (“https://chemdata.nist.gov/” (accessed on 15 April 2025)) and finally normalized by the total metabolites.

### 2.3. Data Analysis

Plant growth parameters and total nitrogen data were analyzed by ANOVA test followed by Tukey’s HSD (honest significant difference) test at *p* < 0.05 level using the R-software (R Core Team 2016). Results were the means ± SD (*n* = 3 or 5). GC-MS data were normalized to tissue dry weight and internal standard. Comparisons between leaf positions and cultivars were performed using the Wilcoxon rank sum test. To visualize metabolites changing across nitrogen stress levels, normalization to the control and log-transformed data were performed in Microsoft Excel 5.0.

## 3. Results and Discussion

### 3.1. Growth Parameters and Total Nitrogen Content of Maize Seedlings Under Different Nitrogen Stress Levels

Different supply levels of nitrogen had a significant influence on maize growth parameters across two cultivars, MB and TYC (Figure 1). As nitrogen levels increased, plant height (PH), shoot fresh weight (SFW), and shoot dry weight (SDW) generally showed an upward trend, particularly between N0 and N80–N100, indicating improved vegetative growth under higher nitrogen availability (Figure 1B–D). In both cultivars, the lowest values were consistently observed under nitrogen-deficient conditions (N0), while maximal growth was achieved at N100, beyond which the response plateaued or slightly declined. Compared to the control (N100), seedlings under N0 reduced 11–14% plant height, and over 30% in shoot fresh weight. Total nitrogen content in both lower leaves (LL) and upper leaves (UL) also increased with rising nitrogen supply, with significant differences observed among treatments (Figure 1E,F). The low-N treatments, N0 and N20, showed the lowest total N content, accounting for less than half of the nitrogen level observed under the control treatment, whereas higher nitrogen levels (N80–N120) resulted in significantly greater leaf N concentration, particularly in upper leaves. This suggests that a sufficient nitrogen supply not only enhances biomass production but also facilitates nitrogen assimilation into plant tissues, which is crucial for photosynthesis and overall plant health.

The two-way ANOVA analysis showed that nitrogen treatment had a significant effect on all traits measured (Table 2). Cultivar effects were significant for SFW and TN, while leaf group effects were particularly significant for TN, indicating the allocation of nitrogen among leaves. Interaction effects (e.g., Cultivar × Treatment, Treatment × Leaf group) were significant for TN, suggesting that nitrogen content is influenced by the combination of genotype, treatment level, and leaf position. Based on consistent trends across growth parameters and cultivars, N0, N60, and N100 were selected to represent deficiency, moderate, and sufficient nitrogen conditions for subsequent metabolite analysis.

Mechanisms by which maize seedlings respond to insufficient N were recorded as early as the initial growth stages [26]. Root NO_3_− uptake capacity and the expression of NO_3_− transporter genes were upregulated from 12 days after imbibition, as an adaptive response to meet the plant’s N demand. Many studies reported growth reduction under N stress in maize seedlings [7,8,9,10,11]. N deprivation caused a reduction in shoot biomass and leaf growth in the seedlings [7,9,29], as well as in stem height and root fresh weight [8]. Consistent with these findings, similar trends were observed in this study. However, the magnitude of these responses may also be influenced by treatment duration and growth period. For instance, the reduction in shoot biomass and leaf growth rate was significantly higher in 30-day-old seedlings than in 20-day-old seedlings [29]. In this study, significant differences in plant growth were primarily observed between N-starved (N0) and N-sufficient (N80–N100) treatments. This limited response range may suggest that the 15-day treatment duration was not sufficient to elicit more pronounced physiological differences across all nitrogen levels. Additionally, before the treatment period, seedlings were maintained for two weeks in a half-strength nutrient solution (equivalent to N50), which may have allowed plants to accumulate and store nitrogen reserves, partially buffering the effects of subsequent nitrogen deficiency and delaying the onset of more severe symptoms. Early-stage N stress can have a negative impact on later developmental processes, including grain filling. A delay in N supply to the six-leaf stage led to appreciate 12% loss in grain yield when the SI (sufficiency index) fell below 0.90, indicating N deficiency can be severe enough to inhibit full yield recovery even with side-dressed N application [30].

### 3.2. GC-MS Metabolite Changes in Maize Seedlings Under Nitrogen Deficiencies

Changes in the central metabolites in corn seedling leaves were normalized to the control treatment (N100) as a reference point in terms of easy comparisons between 2 cultivars and between 2 leaf groups. Therein, the GC-MS data were normalized to the value of N100, then log 2 transformed; thus, the relative changes in reduced-N treatments to the control were expressed as log2 [fold change] as illustrated in Figure 2. Overall, nitrogen deficiency induced distinct shifts in primary metabolite profiles in maize seedlings, with clear differences observed between treatments, leaf positions, and cultivars (Figure 3, Appendix A). Over 25 metabolites were identified, only 6 of which exhibited statistical differences between the 2 cultivars, MB and TYC, while most showed strong differences between upper leaves (UL) and lower leaves (LL) (Appendix A).

Carbohydrate metabolites, including glucose (Glc), fructose (Fru), galactose (Gal), and mannose (Man), consistently accumulated (red shading) under nitrogen-deficient conditions, indicating a carbon build-up resulting from suppressed nitrogen assimilation (Figure 2). In MB, the strongest accumulation was observed for fructose and mannose levels in the lower leaf group, with levels increasing up to nearly threefold. In contrast, TYC showed a greater magnitude in mannose levels under N60 treatment. Galactose exhibited a relatively modest increase, while sucrose (Suc) displayed inverse trends between the two cultivars, slightly decreasing in MB leaves and slightly increasing in TYC leaves. Additionally, statistical tests revealed that sugar contents in LL groups were significantly higher than those in UL groups (Appendix A), except for sucrose, suggesting a differential carbon allocation pattern between leaf positions under nitrogen-deficient conditions.

As expected, under nitrogen-deficient conditions, most amino acids (AAs) exhibited significant reductions (blue shading), which might result from the reduction in total nitrogen content (Figure 2). Glycerate-derived amino acids (glycerin, serine) and aromatic amino acids (tyrosine, phenylalanine) showed decreased relative abundance in both cultivars. In glycerin (Gly), the greater decrease extent was observed under N-starvation treatment, while for other amino acids, the abundant levels were generally higher in UL compared to LL. The same trends were observed in amino acids derived from pyruvate (alanine, valine, leucine), further suggesting that nitrogen limitation affects multiple biosynthetic pathways. Key nitrogen-related amino acids derived from the TCA cycle, such as glutamine (Gln), glutamate (Glu), asparagine (Asn), and aspartate (Asp), were strongly downregulated (up to five-fold) in both leaf positions. However, threonine (Thr) and proline (Pro) deviated from the general pattern, which showed unchanged to slightly increased. The extent of reduction varied between cultivars and leaf positions, with MB showing slightly higher reductions in lower leaves than TYC (Appendix A).

Unlike organic acids displayed mixed responses, as seen in quinic acid (QA) and shikimic acid (SA) (Figure 2). Regarding the TCA-cycle-related organic acids, malic acid (MA) was slightly reduced under nitrogen deficiency except for LL in MB, while citric acid (CA) showed less variation, indicating altered carbon flux through central metabolism. Except for shikimic acid, other organic acids exhibited greater reductions in upper leaf groups (Appendix A), a trend that is consistent with the pattern observed for most amino acids.

Subsequent principal component analysis (PCA) was used to evaluate the primary metabolite responses of maize seedlings to nitrogen availability in two cultivars, Mibeak#2 (MB) and Taeyangchal (TYC), across upper (UL) and lower (LL) leaf tissues (Figure 3). Accordingly, the two cultivars showed very similar trends of PCA projection, indicating a general pattern of changes in central metabolism. In both genotypes, N-treatment groups were clearly separated along the first principal component (PC1), which explained 56.6% of the total variance in MB and 53.1% in TYC, indicating that nitrogen availability was the dominant factor influencing metabolite profiles. PCA loading analysis confirmed that sugars and organic acids were the primary drivers of variation along PC1 under nitrogen-deficient conditions, while amino acids contributed strongly to N-enriched profiles. Nitrogen-starved treatment (N0) was associated with the accumulation of soluble sugars (Glc, Fru, Man, Mal) and organic acids (SA, QA), indicating carbon accumulation and reduced nitrogen assimilation. In contrast, nitrogen-sufficient plants (N100) aligned with elevated levels of nitrogen-containing amino acids such as Glu, Gln, Asp, and branched-chain amino acids (Val, Leu, Ile). In MB, treatment groups showed stronger and clearer separation along the nitrogen gradient, with tightly clustered N100 and N0 groups and a more orderly transition for N60. The second principal component (PC2), explaining 16.5% of variance in MB and 26.7% in TYC, partially separated UL and LL samples, indicating tissue-specific metabolic partitioning. While leaf position had a detectable influence on TYC, tissue separation was less distinct compared to MB. Loadings along PC2 indicated that variation in organic acids (e.g., malate) and aromatic amino acids (e.g., phenylalanine, tyrosine) contributed to tissue-based differentiation.

Collectively, these results suggest that under nitrogen-limited conditions, maize seedlings shift their primary metabolism toward carbon accumulation and nitrogen conservation, with clear cultivar- and tissue-specific differences in metabolic response. These differences may reflect underlying variation in nitrogen use efficiency and stress resilience, thus highlighting the importance of genotype-specific strategies in optimizing nitrogen management for early seedling development.

### 3.3. Effect of Nitrogen Deficiencies on Primary Metabolite Changes

The metabolite changes observed under nitrogen stress conditions were closely aligned with physiological growth reductions and total N content in maize seedlings. The decrease in key amino acids aligns with other research in seedling leaves of maize [9], barley [31], tomato [32], and cabbage [33], reflecting a suppressed nitrogen assimilation pathway, which is essential for protein biosynthesis and cell expansion [34].

However, individual amino acid responses varied under specific conditions. In maize seedling leaves at the V6 stage, many amino acids exhibited coordinated reductions in response to low nitrogen (N) stress, for example, glutamate closely correlated to other major amino acids (Asp, Asn, Ala, Ser, and Gly) [9]. Nevertheless, certain amino acids still deviated from the general trends (tryptophan and lysine), and the extent of reduction varied depending on genotype [9]. Similar patterns were observed in maize leaves grown under field conditions at the same developmental stage, where N-limitation caused a reduction in Phe, Ala, Glu, His, while valine and proline remained unaffected compared to plants supplied with standard N levels [25]. Glutamate serves as a key donor in the biosynthesis of numerous amino acids [35]; therefore, the down-regulation of glutamate under low N availability may contribute to the overall reduction in amino acid pools. Although threonine (Thr) was previously reported to decline in response to N limitation [9], its concentration remained unchanged under both reduced-N treatments in the present study (Figure 2). Nitrogen stress also leads to the accumulation of proline, especially in severe stress plants (N0), which are not primarily used to synthesize protein but function as compatible solutes and as an antioxidant [36,37]. Proline contributes to osmotic adjustment, membrane stabilization, and redox balance functions [4], which are important in stress defense and conservation of energy [37]. However, the magnitude of proline increase was not sufficient to offset the overall stress impact on growth (Figure 1).

The transcriptional mechanisms by which nitrogen deficiency regulates amino acid pools remain incompletely understood. In rice, N deprivation suppressed the transcription of genes involved in amino acid biosynthesis while upregulating those associated with amino acid degradation, which was consistent with their changes in the metabolome [38]. Recent studies have further demonstrated that under nutrient stress conditions, the SNF1-related protein kinase 1 (SnRK1) signaling, a TOR repressor, was activated, thereby promoting amino acid catabolism and shifting the plant’s metabolic focus from growth toward energy conservation and storage [14,39]. These findings suggest that low N availability primarily inhibits the amino acid synthesis process. However, the enzymes participating in amino acid metabolic pathways are regulated by a complex and multilayered mechanism; thus, the individual transcriptional responses to N-deficiencies might not be uniform [40]. In maize seedling leaves under low-N conditions, only asparagine metabolism and Tryptophan biosynthesis could be connected to their transcriptional response, while other pathways showed weaker or no correlation [9].

TCA-related organic acids such as citric acid, malic acid, furmaric acid, succinic acid, were reported to decrease under low-N stress in maize seedlings [9,25], likely reflecting the suppression in central carbon metabolism under nitrogen limitation. Among them, malic acid plays multiple roles in plant metabolism, including functioning in the C_4_ carbon shuttle, pH regulation, carbon and redox balance, and stomatal movement [41]. Therefore, the observed decrease in malate under nitrogen deficiency may reflect a reduced metabolic demand, particularly due to the downregulation of photosynthesis and nitrate assimilation. Similarly, in rice, pathways related to photosynthesis and the pentose phosphate pathway (PPP), two other carbon metabolism pathways, were found to be inhibited under nitrogen deficiency, likely due to a lower demand for NADPH, which is normally required for nitrate reduction [38]. In contrast, organic acids derived from the shikimic pathway were recorded to increase in abundance [9,25]. This accumulation may be linked to upregulation of genes for shikimate biosynthesis precursors, alongside a downregulation of shikimate kinase, the enzyme that converts shikimate into precursors for the aromatic amino acids [9]. In our study, shikimic acid and quinic acid showed mixed responses, suggesting a more nuanced regulation (Figure 2). The shikimate pathway plays an important role in the synthesis of secondary metabolites [1], including aromatic hormones such as auxin and salicylic acid that contribute to plant growth and stress adaptation; therefore, its regulation is likely influenced by factors beyond nitrogen availability alone [42].

Furthermore, the accumulation of soluble sugars such as glucose, fructose, and mannose under N-deficient conditions suggests a metabolic imbalance between carbon and nitrogen. While carbon assimilation through photosynthesis continued, limited nitrogen availability constrained its incorporation into nitrogenous compounds; therefore, carbon is translocated into energy storage compounds like sugar and starch [20,22]. This phenomenon often coincides with reduced sink activity, and its feedback may regulate photosynthetic activity, further compounding growth inhibition. However, the responses of soluble sugar to nitrogen stress may vary with environmental conditions. In a greenhouse experiment with maize seedlings, glucose and fructose levels decreased under low-N conditions [9]. In contrast, fructose and sucrose were significantly increased under field-grown conditions at the same developmental stage [25].

In this study, primary metabolite changes were affected by both tissue positions and genotypes (Figure 3). Sugar (except for sucrose) and organic acids (except for Shikimic acid) were accumulated at higher levels in lower leaves, whereas most of the amino acid pools were more abundant in the upper leaves (Appendix A). These contrasting patterns may reflect developmental and functional gradients along the leaf axis. Lower leaves, being older and more photosynthetically mature, may act as carbon sources, accumulating sugars and organic acids as storage or transport forms. In contrast, upper leaves, typically younger and more actively growing, may function as nutrient sinks, demanding more nitrogen for protein synthesis, which is reflected in the elevated amino acid levels [43]. Interestingly, certain metabolites deviated from these general trends, suggesting pathway-specific regulation beyond simple carbon–nitrogen dynamics. For instance, sucrose export from source to sink leaves is closely associated with floral transition in maize, as it is required to fuel the metabolic requirements of the rapidly growing inflorescence [44]. Moreover, sucrose showed genotype-specific clustering: in MB, it was associated with N100 lower leaves, while in TYC, it clustered near N0. Similarly, shikimic acid displayed opposing trends between genotypes. In MB, it was positioned on the left side of PC1, whereas in TYC, it shifted toward nitrogen-deficient conditions and clustered near the N0 upper leaves. These observations highlight the potential influence of genotype-specific responses and additional regulatory factors, such as stress perception, source–sink reprogramming, or differential enzyme activity, on metabolite distribution within tissues.

## 4. Conclusions and Future Perspective

In conclusion, this study showed that nitrogen availability significantly influences growth and primary metabolism in maize seedlings, with distinct responses observed between genotypes and leaf positions. Nitrogen starvation (N0) led to a reduction in plant height and shoot biomass, further shifting plant metabolism toward carbon-rich compounds, such as soluble sugars and organic acids, whereas nitrogen sufficiency (N100) promoted the accumulation of nitrogen-containing amino acids involved in protein synthesis and growth. Statistical analyses (PCA and ANOVA) revealed that both genotype (MB and TYC) and tissue type (upper and lower leaves) influenced the magnitude and direction of these responses. Notably, MB exhibited more consistent nitrogen-dependent metabolic shifts, while TYC showed greater metabolic variability under mild nitrogen stress. These findings highlight the importance of genotype- and tissue-specific responses in nitrogen availability and suggest that early-stage metabolic profiling can provide valuable insight for improving nitrogen use efficiency in maize cultivation.

To gain a more comprehensive understanding of the metabolic responses to nitrogen deficiency, future studies should consider extending the duration of treatment and incorporating both earlier and later sampling timepoints. This would help capture dynamic shifts in metabolite accumulation and redistribution that may be transient or stage-specific. Moreover, while changes in primary metabolite levels have been documented, there remains a critical need to investigate the underlying transcriptional regulation, not only of biosynthetic pathways but also of key transporters responsible for the mobilization of sugars, amino acids, organic acids, and inorganic nitrogen. Transporter genes play a pivotal role in source–sink communication and nutrient allocation, and their expression may significantly influence the plant’s capacity to adapt to N-limited conditions [45]. Integrating transcriptomics with metabolomics and spatial sampling across developmental stages will be essential for elucidating the complex regulatory networks of carbon and nitrogen metabolism in response to nitrogen availability.

## Figures and Tables

**Figure 1 metabolites-15-00519-f001:**
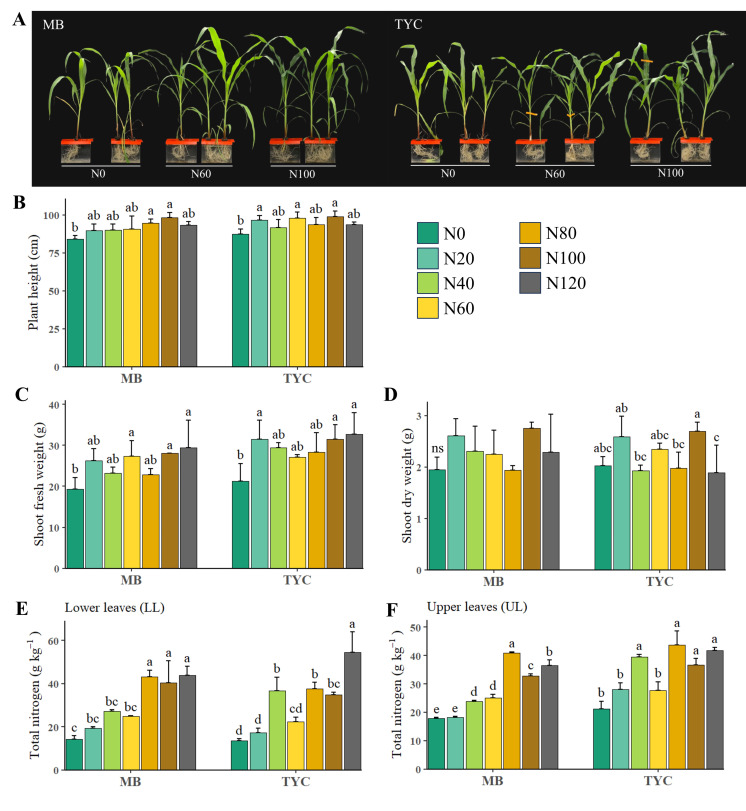
Growth and total nitrogen content of corn seedlings under different nitrogen levels. (**A**) Phenotypes of Mibaek (MB) and Taeyangchal (TYC) seedlings after 15DAT. (**B**) Plant height (cm) of corn seedlings. (**C**) Shoot fresh weight (g) of corn seedlings. (**D**) Shoot dry weight of corn seedlings. Total nitrogen content (g kg^−1^) of corn seedling in lower leaves (**E**) and upper leaves (**F**). Error bars indicate SD. Different letters above the bars indicate statistically significant differences between treatments (*n* = 5, *p* < 0.05, Tukey’s HSD). N0, N20, N40, N60, N80, N100, and N120 correspond to 0%, 20%, 40%, 60%, 80%, 100%, and 120% of the standard nitrogen level (8.5 mM).

**Figure 2 metabolites-15-00519-f002:**
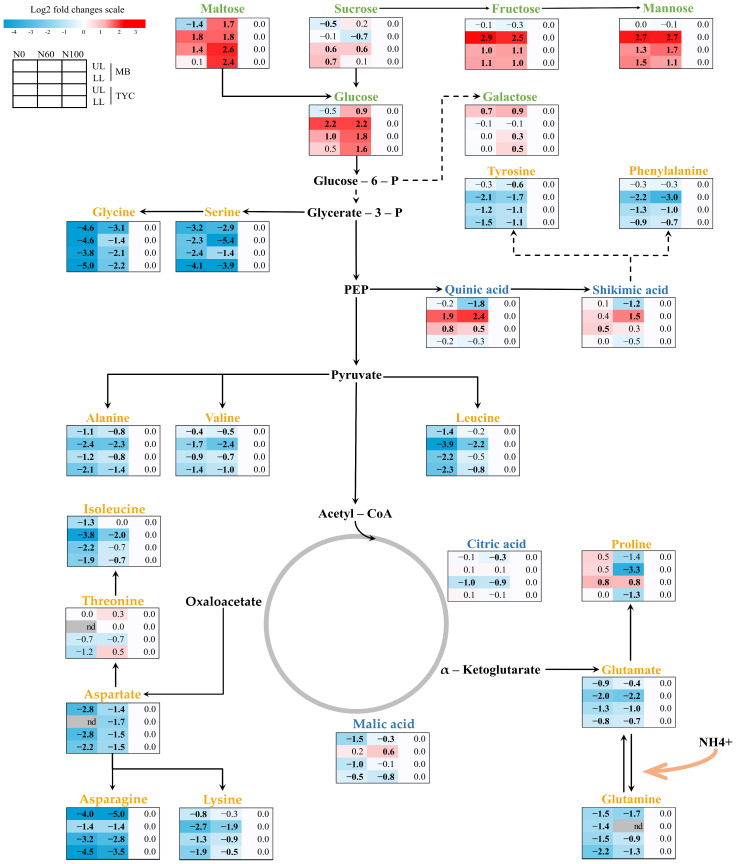
Schematic representation of primary metabolite pathways in maize seedlings with log2 transformed fold change normalized to the control treatment (N100). Red indicates an increase, and blue indicates a decrease in metabolite levels, as shown in the color legend (*n* = 3). Each row shows the changes in metabolites of two leaf groups, upper leaves (UL) and lower leaves (LL), across two cultivars, Mibeak#2 (MB) and Taeyangchal (TYC). Columns represent different nitrogen treatments, with N0 (0% of standard nitrogen supply), N60 (60%), and N100 (100%, used as control). Bold numbers indicate statistically significant differences from N100 based on Tukey’s HSD test (*p* < 0.05). Non-significant comparisons are shown in regular font; see Appendix A for full statistical details. Grey color represents not detected (nd) values.

**Figure 3 metabolites-15-00519-f003:**
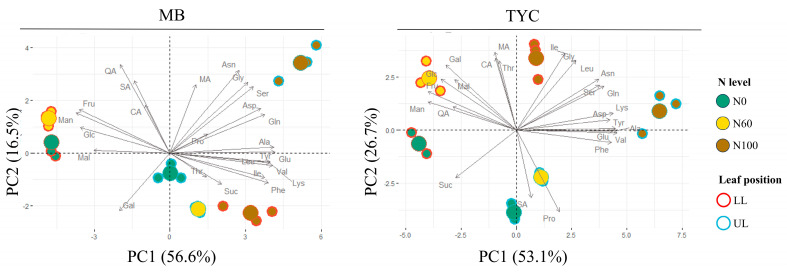
Principal component analysis (PCA) biplots of primary metabolite profiles in maize seedlings of two cultivars (MB and TYC) under three nitrogen levels (N0, N60, N100). Arrows represent metabolite loadings; points represent sample scores colored by N level and shaped by leaf position (UL or LL). Separation along PC1 reflects nitrogen-dependent metabolic shifts, while PC2 captures tissue-specific variation. N levels N0 (0% of standard nitrogen supply), N60 (60%), and N100 (100%).

**Table 1 metabolites-15-00519-t001:** Nitrogen component and equivalent CaCl_2_ and KCl compensation of N-treated solution. The component unit is mM. N0, N20, N40, N60, N80, N100, and N120 correspond to 0%, 20%, 40%, 60%, 80%, 100%, and 120% of the standard nitrogen level (8.5 mM), respectively.

Components	Unit	N0	N20	N40	N60	N80	N100	N120
Ca(No_3_)_2_	mM	0	0.5	1	1.5	2	2.5	3
KNO_3_	mM	0	0.5	1	1.5	2	2.5	3
NH_4_NO_3_	mM	0	0.1	0.2	0.3	0.4	0.5	0.6
CaCl_2_	mM	2.5	2	1.5	1	0.5	0	0
KCl	mM	2.5	2	1.5	1	0.5	0	0
N concentration	mM	0	1.7	3.4	5.1	6.8	8.5	10.2

**Table 2 metabolites-15-00519-t002:** Results of two-way ANOVA for plant growth and total nitrogen content under different nitrogen levels. Factors include cultivar (MB vs. TYC), nitrogen treatment (N0, N60, N100), and leaf group (UL vs. LL). PH, plant height; SFW, shoot fresh weight; SDW, shoot dry weight; TN, total nitrogen content. *p*-values are shown for main effects and interactions; *, significant effects (*p* < 0.05); ns, not significant effects.

ANOVA	PH	SFW	SDW	TN
Cultivar	0.138 (ns)	0.005 (*)	0.624 (ns)	0.0000485 (*)
Treatment	0.00021 (*)	3.80 × 10^−5^ (*)	0.000319 (*)	6.07 × 10^−29^ (*)
Cultivar × Treatment	0.537 (ns)	0.975 (ns)	0.687 (ns)	2.00 × 10^−5^ (*)
Leaf group				0.729 (ns)
Cultivar × Leaf group				4.77 × 10^−4^ (*)
Treatment × Leaf group				8.10 × 10^−6^ (*)
Cultivar × Treatment × Leaf group				0.101 (ns)

## Data Availability

The raw data supporting the conclusions of this article will be made available by the authors on request.

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
