# Peer review of "Primary Metabolic Variations in Maize Plants Affected by Different Levels of Nitrogen Supply"

_metabolites, 2025, doi:10.3390/metabo15080519_

Round 1
Reviewer 1 Report
Comments and Suggestions for Authors
Recommendation: Major Revisions Required
The study addresses a relevant question in plant physiological responses to nitrogen stress and utilizes metabolomics effectively. However, it requires significant improvements in methodology transparency, statistical integration, and deeper mechanistic discussion to reach the standards expected for publication in a Q1 journal. My cpmmnets are as bellow:
Major Comments
- The study explores early-stage nitrogen responses, yet the experimental duration post-treatment (15 days) may be too brief to fully capture systemic metabolic shifts. Early nitrogen buffering from the initial half-strength Hoagland solution may have masked treatment effects. Extending treatment duration or including earlier and later timepoints would enhance clarity and robustness of developmental metabolic trajectories.
- While the study highlights genotype-specific responses (MB vs. TYC), the data presented lacks deeper genetic or physiological context. Without exploring why MB shows more consistent responses, the findings remain observational. Incorporating transcriptomic or transporter expression data could clarify underlying regulatory mechanisms.
- The abstract provides a general summary but omits specific numeric findings or statistical values that substantiate key claims. Including quantified differences (e.g., % reduction in Glu or shoot biomass) would improve scientific precision and better communicate the study's impact.
- The manuscript frequently refers to "nitrogen-related" amino acids, yet does not clearly distinguish between biosynthetic pathways (e.g., glutamate family vs. branched-chain). A more structured metabolic categorization in results/discussion would enhance interpretation and connect better with the nitrogen-carbon flux framework.
- The role of organic acids in nitrogen metabolism is underdeveloped in the discussion. For instance, changes in malate and citrate are only briefly noted. Given their key function in nitrogen assimilation and TCA cycling, their regulation under N stress should be interpreted more mechanistically.
- The PCA biplot analysis is informative but lacks sufficient narrative integration. The text should explicitly discuss the percentage of variance explained, significance of clustering patterns, and loadings of key metabolites in the PCA axes to support interpretations about genotype and tissue differences.
- The rationale for selecting only N0, N60, and N100 for metabolite profiling is insufficiently justified. Intermediate levels such as N20 or N80 could have revealed thresholds or non-linear patterns in metabolic responses. Including more gradient points could refine conclusions about dose-dependent changes.
- The methodology lacks detail regarding biological replication in GC-MS metabolite profiling. Were technical replicates included? How many biological samples per genotype/treatment/tissue were profiled? Clarifying this is critical for assessing data reliability.
- There is an overreliance on referencing prior literature without deeper analytical contrast. For example, while findings are aligned with past reports, the novelty or divergence from earlier work is not critically addressed. A more nuanced comparative analysis would elevate the discussion.
- The figures (especially Figure 2) use color-coding (red/blue) but lack clear legends or statistical annotations. Readers may find it difficult to judge significance or biological relevance of changes. Adding error bars or asterisks for significant differences would improve clarity.
- Although the study investigates both upper and lower leaves, the physiological significance of these positions is not well articulated. Do differences reflect developmental gradients, source-sink dynamics, or differential stress perception? Elaboration would strengthen the tissue-specific argument.
- The N concentration levels (Table 1) do not clarify whether total N concentration adjustments preserved ion balance (especially K, Ca). Imbalances in nutrient composition could confound the interpretation of N-specific effects. This requires elaboration or control experiments.
- The manuscript does not discuss the potential role of other primary metabolic regulators like the TOR signaling pathway beyond citation. If invoked, it would be helpful to mention relevant evidence from the metabolomic or phenotypic data that links to such signaling pathways.
- There is minimal engagement with the implications for agronomic breeding or screening. The study could better highlight how MB’s consistent response could serve as a marker for early nitrogen stress tolerance and how that might inform cultivar selection or management practices.
Minor Comments
- The phrase “tissue-sepcific” in the keywords section contains a typographical error and should be corrected to “tissue-specific” to improve searchability and professionalism.
- “Turkey’s HSD” is misspelled and should be corrected to “Tukey’s HSD” throughout the manuscript for accurate reference to the statistical test.
- The term “unlikely” in “Unlikely, organic acids displayed…” is incorrect; it should be “unlike” for proper grammatical contrast.
- In the abstract, "elicited milder responses" is vague. Replacing it with a more precise phrase like “resulted in moderate reductions in growth and nitrogen content” would be clearer.
- The sentence “The nutrient solution at 15 days after transplanting was replaced…” is awkward; consider rephrasing to “At 15 days after transplanting, the nutrient solution was replaced with treatments differing in nitrogen concentration.”
- The citation style is inconsistent (e.g., [9], see [13] for a review). Ensure all in-text references follow the journal's prescribed citation format.
- In multiple instances, unnecessary phrases like “in another hand” are used. Replace with correct idiomatic expressions like “on the other hand” for smoother flow.
- Figure 3's legend lacks a clear description of what Dim1 and Dim2 represent in terms of explained variance; this should be added for clarity.
- The reference list has inconsistent punctuation and spacing (e.g., reference [2] and [15]). The formatting should be standardized according to journal guidelines.
- The abstract's final sentence should be split into two for better readability. The current compound structure hampers quick understanding of the conclusion and implications.
Comments on the Quality of English Language
The manuscript is generally understandable, but it requires significant improvements to the quality of the English. Several sentences contain grammatical errors, awkward phrasing, and improper word choices that affect clarity and professionalism. Common issues include the incorrect use of transitional expressions (e.g., "in another hand"), inconsistent verb tenses, and run-on sentences. To enhance readability and ensure the scientific message is clearly conveyed, careful proofreading by a native English speaker or a professional editing service is strongly recommended.
Author Response
Thank you very much for taking the time to review this manuscript. We thank for the positive evaluation. Please find the detailed responses below and the corresponding revisions/corrections highlighted/in track changes in the re-submitted.
Major Comments
Comments 1: The study explores early-stage nitrogen responses, yet the experimental duration post-treatment (15 days) may be too brief to fully capture systemic metabolic shifts. Early nitrogen buffering from the initial half-strength Hoagland solution may have masked treatment effects. Extending treatment duration or including earlier and later timepoints would enhance clarity and robustness of developmental metabolic trajectories.
Response 1: We appreciate the reviewer’s valuable comment. We had discussion this point in the section 3.1, and in section 4 (Conclusion and Future Perspective), we also added a statement emphasizing the need for future studies that include earlier and later timepoints to better capture dynamic metabolic responses as following: “To gain a more comprehensive understanding of the metabolic responses to nitrogen deficiency, future studies should consider extending the duration of treatment and incorporating both earlier and later sampling timepoints. This would help capture dynamic shifts in metabolite accumulation and redistribution that may be transient or stage specific.” (Page 11, line 412-415)
Comments 2: While the study highlights genotype-specific responses (MB vs. TYC), the data presented lacks deeper genetic or physiological context. Without exploring why MB shows more consistent responses, the findings remain observational. Incorporating transcriptomic or transporter expression data could clarify underlying regulatory mechanisms.
Response 2: We understand the reviewer’s concern. In this study, we used two Korean waxy corn cultivars with different backgrounds and agronomic relevance. Mibaek#2 (MB), developed in 2005 from inbred lines HW9 and HW3, is one of the most widely cultivated varieties in Korea due to its high yield and resistance to lodging, B.maydis, and E.turcicum. Taeyangchal (TYC), on the other hand, is a newly released variety developed by Chungbuk Province (South Korea). We agree that understanding the regulatory basis behind MB's consistent responses is important. While this current study focused on phenotypic and metabolic traits, we are planning follow-up research incorporating transcriptomic and transporter expression analyses to investigate the underlying mechanisms. However, due to time constraints, these data are not yet available for inclusion in this manuscript.
Comments 3: The abstract provides a general summary but omits specific numeric findings or statistical values that substantiate key claims. Including quantified differences (e.g., % reduction in Glu or shoot biomass) would improve scientific precision and better communicate the study's impact.
Response 3: As suggested by the reviewer, we have revised the abstract (page 1, line 16-18, line 20).
Comments 4: The manuscript frequently refers to "nitrogen-related" amino acids, yet does not clearly distinguish between biosynthetic pathways (e.g., glutamate family vs. branched-chain). A more structured metabolic categorization in results/discussion would enhance interpretation and connect better with the nitrogen-carbon flux framework.
Response 4: We appreciate the reviewer’s suggestion. In response, we have restructured the Results and Discussion sections to categorize amino acids based on their biosynthetic pathways (e.g., glutamate family, branched-chain, aspartate family), which allows for a clearer interpretation of their relationship with nitrogen and carbon metabolism. Details can be found in section 3.2 (page 7, line 240-253) and section 3.3.
Comments 5: The role of organic acids in nitrogen metabolism is underdeveloped in the discussion. For instance, changes in malate and citrate are only briefly noted. Given their key function in nitrogen assimilation and TCA cycling, their regulation under N stress should be interpreted more mechanistically.
Response 5: We revised the Discussion as the reviewer suggested (page 10, line 345-364).
Comments 6: The PCA biplot analysis is informative but lacks sufficient narrative integration. The text should explicitly discuss the percentage of variance explained, significance of clustering patterns, and loadings of key metabolites in the PCA axes to support interpretations about genotype and tissue differences.
Response 6: We appreciate the reviewer’s suggestion. In response, we have revised the text in section 3.2 (pages 7–8, lines 261–282) to more explicitly describe the PCA results, including the percentage of variance explained by the principal components, the clustering patterns among genotypes and tissue types, and the major metabolite loadings on each axis. These additions enhance the interpretative value of the PCA and support our conclusions regarding genotype- and tissue-specific responses.
Comments 7: The rationale for selecting only N0, N60, and N100 for metabolite profiling is insufficiently justified. Intermediate levels such as N20 or N80 could have revealed thresholds or non-linear patterns in metabolic responses. Including more gradient points could refine conclusions about dose-dependent changes.
Response 7: We understand the reviewer’s concern. Our decision to select only N0, N60 and N100 was based on their consistent and distinguishable patterns observed across all growth parameters and in both cultivars (as shown in Figure 1). Specifically, N0 consistently represented the most severe nitrogen deficiency with significantly reduced growth and nitrogen content, while N100 showed the highest values, serving as the control for sufficient nitrogen. N60 represented a moderate N-level that marked the transition zone, where shoot biomass and total nitrogen content showed intermediate values.
In contrast, N20 and N40 produced similar responses to N0 and N60, respectively, with no statistically significant differences in many parameters (Figure 1B–F). Likewise, N80 and N120 clustered closely with N100, indicating saturation effects at high N levels. Therefore, including more treatments in metabolite analysis would have added little additional insight in this experiment. However, we agree that including more levels could refine the understanding of dose-dependent effects, and we plan to explore this in future studies.
To improve clarity for readers, we add the following explanation in the text: “Based on consistent trends across growth parameters and cultivars, N0, N60, and N100 were selected to represent deficiency, moderate, and sufficient nitrogen conditions for subsequent metabolite analysis.” (page 4-5, line 179-181)
Comments 8: The methodology lacks detail regarding biological replication in GC-MS metabolite profiling. Were technical replicates included? How many biological samples per genotype/treatment/tissue were profiled? Clarifying this is critical for assessing data reliability.
Response 8: As suggested by the reviewer, we added the information about the number of biological samples per treatment in the text (page 3, line 129-130).
Comments 9: There is an overreliance on referencing prior literature without deeper analytical contrast. For example, while findings are aligned with past reports, the novelty or divergence from earlier work is not critically addressed. A more nuanced comparative analysis would elevate the discussion.
Response 9: Thank you for pointing this out. We have revised the discussion in Section 3.3 (page 9-10, line 306-375) to provide a more critical comparison with previous studies, highlighting both consistent and novel aspects of our findings.
Comments 10: The figures (especially Figure 2) use color-coding (red/blue) but lack clear legends or statistical annotations. Readers may find it difficult to judge significance or biological relevance of changes. Adding error bars or asterisks for significant differences would improve clarity.
Response 10: We agree with this comment. We have modified Figure 2 as suggested (page 8, line 289). Statistical analysis was performed using ANOVA followed by Tukey’s HSD test to identify significant differences between nitrogen treatment groups within each leaf group × cultivar combination. To avoid overcrowding in the figure, we didn’t add error bars or asterisks; instead, we used bold font to indicate statistically significant differences from N100 based on Tukey’s HSD test (p < 0.05). Non-significant comparisons are shown in regular font; full statistical details can be found in Supplementary Table 2.
Comments 11: Although the study investigates both upper and lower leaves, the physiological significance of these positions is not well articulated. Do differences reflect developmental gradients, source-sink dynamics, or differential stress perception? Elaboration would strengthen the tissue-specific argument.
Response 11: Thank you for the valuable comment. We have revised the discussion in Section 3.3 (page 10–11, lines 376–396) to better elaborate on the physiological significance of upper vs. lower leaves.
Comments 12: The N concentration levels (Table 1) do not clarify whether total N concentration adjustments preserved ion balance (especially K, Ca). Imbalances in nutrient composition could confound the interpretation of N-specific effects. This requires elaboration or control experiments.
Response 12: We added information about equivalent CaCl2 and KCl compensation in Table 1 (Page 3, line 127)
Comments 13: The manuscript does not discuss the potential role of other primary metabolic regulators like the TOR signaling pathway beyond citation. If invoked, it would be helpful to mention relevant evidence from the metabolomic or phenotypic data that links to such signaling pathways.
Response 13: We have revised Section 3.3 as suggested.
Comments 14: There is minimal engagement with the implications for agronomic breeding or screening. The study could better highlight how MB’s consistent response could serve as a marker for early nitrogen stress tolerance and how that might inform cultivar selection or management practices.
Response 14: Thank you for the suggestion; however, we have removed this point from the manuscript to maintain focus and avoid overextension beyond the scope of the current data.
Minor Comments
Comments 15: The phrase “tissue-sepcific” in the keywords section contains a typographical error and should be corrected to “tissue-specific” to improve searchability and professionalism.
Response 15: Thank you for pointing this out. We revise the text as reviewer suggested (page 1, line 31)
Comments 16: “Turkey’s HSD” is misspelled and should be corrected to “Tukey’s HSD” throughout the manuscript for accurate reference to the statistical test.
Response 16: As suggested by the reviewer, we have revised the text (page 4, line 149; page 6, line 215).
Comments 17: The term “unlikely” in “Unlikely, organic acids displayed…” is incorrect; it should be “unlike” for proper grammatical contrast.
Response 17: As suggested by the reviewer, we have revised the text (page 7, line 254).
Comments 18: In the abstract, "elicited milder responses" is vague. Replacing it with a more precise phrase like “resulted in moderate reductions in growth and nitrogen content” would be clearer.
Response 18: As suggested by the reviewer, we have revised the text (page 1, line 18-19).
Comments 19: The sentence “The nutrient solution at 15 days after transplanting was replaced…” is awkward; consider rephrasing to “At 15 days after transplanting, the nutrient solution was replaced with treatments differing in nitrogen concentration.”
Response 19: As suggested by the reviewer, we have revised the text (page 3, line 105-107).
Comments 20: The citation style is inconsistent (e.g., [9], see [13] for a review). Ensure all in-text references follow the journal's prescribed citation format.
Response 20: As suggested by the reviewer, we have revised the text (page 2, line 53).
Comments 21: In multiple instances, unnecessary phrases like “in another hand” are used. Replace with correct idiomatic expressions like “on the other hand” for smoother flow.
Response 21: As suggested by the reviewer, we have revised the text (page 2, line 66).
Comments 22: Figure 3's legend lacks a clear description of what Dim1 and Dim2 represent in terms of explained variance; this should be added for clarity.
Response 22: We have revised the figure legend to replace "Dim1" and "Dim2" with "PC1" and "PC2," and added a detailed description (page 9, line 303-304).
Comments 23: The reference list has inconsistent punctuation and spacing (e.g., reference [2] and [15]). The formatting should be standardized according to journal guidelines.
Response 23: The reference list is made following the journal guidelines; the inconsistent space might result from justified texts. Minor corrections were made as suggested.
Comments 24: The abstract's final sentence should be split into two for better readability. The current compound structure hampers quick understanding of the conclusion and implications.
Response 24: As suggested by the reviewer, we have revised the text (page 1, line 25-31).
Reviewer 2 Report
Comments and Suggestions for Authors
Dear Authors, your manuscript of the article provided to me by the editors is a very interesting scientific study. Your research results are very interesting from both a scientific and practical point of view. As a reviewer, I have an obligation to look critically at the manuscript and identify the weak points of the work before its publication. Therefore, I present below in points my questions and suggestions for improving the manuscript. 1. My first reservation concerns keywords. The words nitrogen; primary metabolite; maize - coincide with the words in the title. Keyword is the second search criterion in scientific search engines apart from the title, therefore, according to the generally accepted standard, they must be consistent with the subject of the work but not be in the title. I propose replacing them with the words: nitrogen assimilation; nutrient solution, maize genotype, nitrogen use efficiency; maize cultivation; shoot biomass.
2. I have a question about the varieties. Can you give a more detailed description and explain why you used these varieties in your experiment? Why only two and not more? Are these varieties with some extremely opposite characteristics or are they the most popular among farmers in Korea?
3. Why in section 2.2 "Depending on growth and total nitrogen content, the selected treatments, N0, N60, and N100, were taken for metabolite measurement. - did you select only these nitrogen doses for analysis, not all of them, and omitted 80 and 120? Can you explain this?
4. In general, you have presented and described your research results very correctly. However, I have reservations about the discussion. It should have been more in-depth. In addition, you have used little scientific literature in the discussion, so you have not sufficiently confronted your own research results with the results of other researchers. It is worth supplementing this and using more scientific literature. In your discussion, you have used only 9 items of literature. There is such a wide range of available literature in this field that you should have used much more literature. In my opinion, at least 30 items in the discussion.
Author Response
Thank you very much for taking the time to review this manuscript. We thank for the positive evaluation. Please find the detailed responses below and the corresponding revisions/corrections highlighted in the re-submitted.
Comments 1: My first reservation concerns keywords. The words nitrogen; primary metabolite; maize - coincide with the words in the title. Keyword is the second search criterion in scientific search engines apart from the title, therefore, according to the generally accepted standard, they must be consistent with the subject of the work but not be in the title. I propose replacing them with the words: nitrogen assimilation; nutrient solution, maize genotype, nitrogen use efficiency; maize cultivation; shoot biomass.
Response 1: Thank you for pointing this out. We agree with this comment. We revised the text as reviewer suggested (page 1, line 31).
Comments 2: I have a question about the varieties. Can you give a more detailed description and explain why you used these varieties in your experiment? Why only two and not more? Are these varieties with some extremely opposite characteristics or are they the most popular among farmers in Korea?
Response 2: Thank you for your question. In this study, we used two Korean waxy corn cultivars with different backgrounds and agronomic relevance. Mibaek#2 (MB), developed in 2005 from inbred lines HW9 and HW3, is one of the most widely cultivated varieties in Korea due to its high yield and resistance to lodging, B.maydis, and E.turcicum. Taeyangchal (TYC), on the other hand, is a newly released variety developed by Chungbuk Province (South Korea). We added the short cultivars’ information in the manuscript as following: “MB, developed from 2005, is one of the most popular varieties in Korea, while TYC is a newly released variety in 2022.” (page 3, line 96-97).
Comments 3: Why in section 2.2 "Depending on growth and total nitrogen content, the selected treatments, N0, N60, and N100, were taken for metabolite measurement. - did you select only these nitrogen doses for analysis, not all of them, and omitted 80 and 120? Can you explain this?
Response 3: Thank you for your question. Our decision to select only N0, N60, and N100 was based on their consistent and distinguishable patterns observed across all growth parameters and in both cultivars (as shown in Figure 1). Specifically, N0 consistently represented the most severe nitrogen deficiency with significantly reduced growth and nitrogen content, while N100 showed the highest values, serving as the control for sufficient nitrogen. N60 represented a moderate N-level that marked the transition zone, where shoot biomass and total nitrogen content showed intermediate values.
In contrast, N20 and N40 produced similar responses to N0 and N60, respectively, with no statistically significant differences in many parameters (Figure 1B–F). Likewise, N80 and N120 clustered closely with N100, indicating saturation effects at high N levels. Therefore, including more treatments in metabolite analysis would have added little additional insight in this experiment. However, we agree that including more levels could refine the understanding of dose-dependent effects, and we plan to explore this in future studies.
To improve clarity for readers, we add the following explanation in the text: “Based on consistent trends across growth parameters and cultivars, N0, N60, and N100 were selected to represent deficiency, moderate, and sufficient nitrogen conditions for subsequent metabolite analysis.” (page 4-5, line 179-181)
Comments 4: In general, you have presented and described your research results very correctly. However, I have reservations about the discussion. It should have been more in-depth. In addition, you have used little scientific literature in the discussion, so you have not sufficiently confronted your own research results with the results of other researchers. It is worth supplementing this and using more scientific literature. In your discussion, you have used only 9 items of literature. There is such a wide range of available literature in this field that you should have used much more literature. In my opinion, at least 30 items in the discussion.
Response 4: We appreciate the reviewer’s suggestion. In response, we have thoroughly revised and expanded the Discussion section to provide a more in-depth interpretation of our findings (section 3.3).
Reviewer 3 Report
Comments and Suggestions for Authors
Comments and Suggestions for Authors
Title: Primary metabolic variations in upper and lower leaves of maize plants
under different levels of nitrogen supply
Dear Authors and Editors
The research results presented in the manuscript fall within the publishing profile of the journal Metabolites. The research topic is original and relevant to the field of agricultural sciences.
It's a pity that the research wasn't conducted in field conditions. Therefore, the results should be considered preliminary and cannot be applied to studies conducted under natural soil conditions. The aim of the presented research was to determine the effect of different nitrogen levels on the growth and changes in basic metabolites in maize seedlings, with particular emphasis on the content of amino acids, organic acids and carbohydrates.
Manuscript contains many valuable research results. The manuscript is well planned and presented. Results and Discussion section is well prepared. The conclusions are consistent with the presented research results. All references are used appropriately.
In order to increase the usefulness of the article, Authors must refer to the following points. Additions should be made to increase the scientific value of the manuscript.
Comments are given below.
Comments
- Abstract: Background/Objectives… (L. 9),… Methods… (L. 13), Results (L. 15) and …Conclusions… (L. 23) – should be removed.
- Keywords: …pirimary metabolite; tissue-secific…. – should be changed or clarified.
- Materials and Methods: Subsection 2.1. Please add the mass of the artificial soil in the tray. Please also include the basic physicochemical properties of the artificial soil. Line 110 Why were plant samples dried at 80oC? What methodology was used [source].
- Results and Discussion: Line 174 The number of one-time citations should be reduced. Figure1E and 1F - The total nitrogen content should be given in g kg-1m. Line 226 – Please include Supplement Table 1 in subsection 3.2.
- Conclusions: Do the authors see a need for further research on this topic? If so, please outline directions for further research.
Specific comments
- Line 51…see…; Line 52….for a review…. – should be removed.
- References: Minor corrections should be made in accordance with publishing requirements.
Best regards
Author Response
Thank you very much for taking the time to review this manuscript. We thank for the positive evaluation. Please find the detailed responses below and the corresponding revisions/corrections highlighted in the re-submitted.
Comments 1: Background/Objectives… (L. 9),… Methods… (L. 13), Results (L. 15) and …Conclusions… (L. 23) – should be removed.
Response 1: We appreciate the reviewer’s suggestion. However, these section headings (Background/Objectives, Methods, Results, and Conclusions) are part of the journal’s required abstract formatting and cannot be removed. To improve readability and avoid redundancy, we have made minor revisions to the text under each heading.
Comments 2: Keywords: …pirimary metabolite; tissue-secific…. – should be changed or clarified.
Response 2: Thank you for pointing this out. We agree with this comment. We revised the text as reviewer suggested (page 1, line 31).
Comments 3: Materials and Methods: Subsection 2.1. Please add the mass of the artificial soil in the tray. Please also include the basic physicochemical properties of the artificial soil. Line 110 Why were plant samples dried at 80oC? What methodology was used [source].
Response 3: We appreciate the reviewer’s suggestion. As stated in Section 2.1 (page 3, lines 100–105), we initially used artificial soil during seed sowing. When the seedlings reached the 2-leaf stage, the roots were gently washed with water to remove any residual soil, and the plants were then transferred to hydroponic containers with half-strength Hoagland nutrient solution. The soil’s influence on the final measurements is expected to be minimal; therefore, we didn’t include this in the manuscript. To address potential concerns and improve clarity, we have now added the phrase “were gently washed with water to remove the soil” in the revised manuscript (page 3, line 101).
Regarding the drying temperature, plant samples were dried at 80 °C for 72h to ensure consistent weight loss to a constant mass, as per standard drying procedures. The appropriate citation has been added to the revised manuscript (page 3, line 117).
Comments 4: Results and Discussion: Line 174 The number of one-time citations should be reduced. Figure 1E and 1F - The total nitrogen content should be given in g kg-1. Line 226 – Please include Supplement Table 1 in subsection 3.2.
Response 4: Thank you for the suggestions. We have revised the unit of total nitrogen content in Figures 1E and 1F to g kg⁻¹ as recommended. Regarding the inclusion of Supplementary Table 1 in subsection 3.2, we believe this may be redundant, as the relevant data are already summarized and discussed in the main text. However, we have ensured appropriate referencing to the supplementary table throughout the manuscript.
Comments 5: Conclusions: Do the authors see a need for further research on this topic? If so, please outline directions for further research.
Response 5: As suggested by the reviewer, we have added the need for further research in section 4 (Conclusion and Future Perspective). The following is our idea: “To gain a more comprehensive understanding of the metabolic responses to nitrogen deficiency, future studies should consider extending the duration of treatment and incorporating both earlier and later sampling timepoints. This would help capture dynamic shifts in metabolite accumulation and redistribution that may be transient or stage-specific. Moreover, while changes in primary metabolite levels have been documented, there remains a critical need to investigate the underlying transcriptional regulation, not only of biosynthetic pathways but also of key transporters responsible for the mobilization of sugars, amino acids, organic acids, and inorganic nitrogen. Transporter genes play a pivotal role in source–sink communication and nutrient allocation, and their expression may significantly influence the plant’s capacity to adapt to N-limited conditions [45]. Integrating transcriptomics with metabolomics and spatial sampling across developmental stages will be essential for elucidating the complex regulatory networks of carbon and nitrogen metabolism in response to nitrogen availability.” (page 11, line 412-424).
Comments 6: Line 51…see…; Line 52….for a review…. – should be removed.
Response 6: We agree with this comment. We revised the text as reviewer suggested (page 2, line 53).
Comments 7: References: Minor corrections should be made in accordance with publishing requirements.
Response 7: As suggested by the reviewer, we have revised the references to comply with the journal's formatting requirements.
Round 2
Reviewer 1 Report
Comments and Suggestions for Authors
I fully endorse the publication of this manuscript. The revised version shows significant improvement in clarity, structure, and data interpretation. I have no further concerns, and the study is now suitable for publication in its current form.